# Effect of Graphene Oxide and Reduced Graphene Oxide on the Properties of Sunflower Oil-Based Polyurethane Films

**DOI:** 10.3390/polym14224974

**Published:** 2022-11-17

**Authors:** Vishwa Suthar, Magdalene A. Asare, Felipe M. de Souza, Ram K. Gupta

**Affiliations:** 1Department of Chemistry, Pittsburg State University, 1701 S. Broadway Street, Pittsburg, KS 66762, USA; 2National Institute for Materials Advancement, Pittsburg State University, 1204 Research Road, Pittsburg, KS 66762, USA

**Keywords:** sunflower oil, bio-based polyurethane, graphene oxide, reduced graphene oxide, composite coating

## Abstract

Sunflower oil was used for the synthesis of a polyol via an epoxidation reaction followed by a ring-opening reaction. The successful synthesis of the sunflower oil-based polyol (SFO polyol) was demonstrated through structural characterizations and wet-chemistry analysis. Bio-based polyurethane (BPU) films were fabricated using synthesized polyol and diisocyanate. Various amounts of graphene oxide (GO) and reduced graphene oxide (rGO) were added separately to see their effect on the physicomechanical and thermal properties of BPU films. Several tests, such as thermogravimetric analysis, tensile strength, dynamic mechanical analysis, hardness, flexural strength, and the water contact angle, were performed to evaluate the effect of GO and rGO on the properties of the BPU films. Some of the analyses of the BPU films demonstrated an improvement in the mechanical properties, for example, the tensile strength increased from 22.5 to 26 MPa with the addition of only 0.05 wt.% GO. The storage modulus improved from 900 to 1000 and 1700 MPa after the addition of 0.02 and 0.05 wt.% GO, respectively. This study shows that a small amount of GO and rGO could improve the properties of BPU films, making them suitable for use in coating industries.

## 1. Introduction

The chemical versatility of reagents and the processing of polyurethanes (PUs) made them highly commercially successful. PUs are used in many applications, such as in protective coatings, dispersions, and elastomers, which can be employed in the automobile industry, surfaces protection, footwear, and construction, among many others. High-performance requirements in these industries are constantly pushing the development and incorporation of novel materials to make them more competitive. PUs can be obtained through facile methodologies that involve a polyaddition reaction between a starting material containing two or more hydroxyl groups, i.e., diol or polyol (–OH), and a polyisocyanate (–NCO) to yield a urethane linkage. In this sense, PUs’ properties are highly influenced by the chemical structure of the polyols, as an aliphatic alkyl chain can introduce more flexibility to the polymeric chain. On the other hand, isocyanate tends to present an aromatic structure and a shorter chain, which induces more rigidity to the polymer backbone. Other factors, such as functionality, branching, and molecular weight also play an important role in PUs’ properties [1], and thus, a wide range of properties can be covered based on the chemical structure of the starting materials. There is also a plethora of available starting materials that allow for versatile properties and applications of PUs.

The introduction of bio-based materials can reduce the pressure on the exhaustive use of petrochemical-based raw materials. Such approaches can lead to the more sustainable production of PU-based products, which can help the environment as well as diversify the sources of starting materials. Several bio-based materials are used for the synthesis of polyols, such as vegetable oils [2,3,4,5,6,7,8,9], essential oils [10,11,12], fruit peels [13,14], and lignin [15,16,17,18], among others. In addition to the versatility of starting materials, several approaches can be used to convert these bio-based materials into polyols, such as hydroformylation [19], thiol–ene coupling [20,21,22,23,24], epoxidation/ring-opening [25,26], etc. For example, Lu et al. [27] synthesized a soybean oil-based polyol through an epoxidation/ring-opening process for the development of waterborne PU dispersion. The control over the functionality of the soybean oil-based polyol yielded films with a wide range of tensile strength (range from 4.2 to 21.5 MPa), along with appreciable elastic behavior, by presenting elongation at break up to 280%. Several other bio-materials can be also chemically modified to make them suitable for polymerization by utilizing their unsaturation as reaction sites [28]. From this perspective, sunflower oil was selected as a feasible bio-renewable option, as it has presented an increasing trend in production over the years, with an estimated average of around 17.75 million metric tons. With this increasing trend of production, sunflower oil is the fourth most consumed oil worldwide, being a relatively abundant resource [29].

Alongside finding renewable sources to obtain materials with comparable properties to those of non-renewable origins, it is also important to incorporate fillers that can promote an enhancement in target aspects, such as thermal stability, mechanical properties, flexibility, and chemical resistance. Ideally, satisfactory improvements should be achieved with lower additions of fillers, which can lead to cost savings as well as better sustainable credentials. Both GO and rGO are suitable options to attend to such requirements due to their inherent properties related to high flexibility, as well as mechanical strength and thermal and chemical stability [30,31,32,33,34,35,36]. Based on these inherent characteristics of GO and rGO, for instance, the properties of PU-based films can be further enhanced by the addition of fillers based on carbon nanomaterials. For example, Deka et al. [37] incorporated multi-walled carbon nanotubes (MWCNTs) into a *Mesua ferrea* L. seed oil-based hyperbranched PU. The composite acquired appreciable properties, such as an introduced self-healing capability, along with an improvement of around 300% in tensile strength. Chen et al. [38] reviewed the effect of the addition of clay into PU dispersion and studied the effect of clay on the properties of PUs. Furthermore, Qian et al. [39] fabricated a composite film based on MWCNT that was homogeneously dispersed in a polystyrene matrix through solvent evaporation followed by a high-energy sonication process. The incorporation of 1 wt.% of MWCNT provided a satisfactory improvement in the mechanical properties, as the break stress and elastic modulus improved around 25% and 36–42%, respectively. Based on these results, it could be noticed that a relatively low amount of CNTs in a polymeric matrix can provide a considerable improvement in properties. However, the inherently low dispersibility of carbon-based materials in polymeric matrices is one of the main challenges, as it leads to agglomeration and phase separation [40,41,42]. Hence, enabling proper ways to improve the dispersibility of these carbon-based nanomaterials can drastically influence the composite’s properties. One of the solutions is to use surface-functionalized carbons. For example, graphene oxide (GO) could improve dispersion in polar solvents, as it contains oxygenated groups in its structure. In addition, GO possesses a larger interlayer spacing, which prevents agglomeration [43,44]. Reduced GO (rGO) is another carbon-based nanomaterial that contains a reduced number of oxygenated functions and can be used as a filler [41,45,46].

Based on these studies, this work consists of the fabrication of a PU composite coating synthesized from a sunflower oil (SFO)-based polyol, which was synthesized via the epoxidation/ring-opening route. During the film formulation process, GO and rGO were added in increasing concentrations to fabricate composite coatings (Figure 1). The effects of the concentrations of GO and rGO were studied. This work describes a reproducible and versatile approach for the production of thin bio-based PU films, which can be obtained through the use of virtually any unsaturated vegetable oil. In addition, it shows that lower concentrations of carbon-based nanomaterials, such as GO and rGO can lead to an appreciable increase in properties, mostly mechanical, even when added under lower concentrations. Based on this, we expect that the discussions provided in this work may aid other scientists to incorporate some of the strategies in their future research. Furthermore, the results obtained in this work suggest that these bio-based composite PU films can be potentially applied as a protective coating against mechanical impact, piercing, and moisture.

## 2. Experimental Details

### 2.1. Materials

Rubinate M isocyanate (methylene diphenyl diisocyanate, MDI), was gifted from Huntsman (The Woodlands, TX, USA). Sunflower oil (food grade) and distilled water were purchased from a local Walmart (Pittsburg, KS, USA). Amberlite IR 120H resin, 30 wt.% hydrogen peroxide, sodium chloride (99.0%), ethyl acetate (99.9%), acetic acid (99.7%), methanol (99.9%), anhydrous sodium sulfate (99.0%), and Lewitt MP64 resin were purchased from Fisher Scientific (Fair Lawn, NJ, USA). GO and rGO were purchased from Sigma-Aldrich (St. Louis, MO, USA) and ACS Material (Pasadena, CA, USA), respectively.

### 2.2. Synthesis of Epoxidized Sunflower Oil

The polyol was synthesized using a two-step process consisting of epoxidation followed by ring-opening reactions. In the epoxidation of the sunflower oil, the molar ratio of double bond/acetic acid/hydrogen peroxide was 1:0.5:1.5. In a three-necked round-bottom flask, 300 g of SFO and 150 mL of toluene (50 wt.% of SFO) were added. Then, 75 g of Amberlite IR 120H resin (catalysts) was added. The mixture was stirred for 15 min until homogeneity was achieved. After that, 43.9 mL of acetic acid was added slowly with a dropping funnel, followed by 30 min stirring. 180 mL of 30% wt./wt. of an H_2_O_2_ solution was added in the same manner while maintaining the temperature between 5 and 10 °C. After the complete addition of H_2_O_2_, the temperature was increased and maintained at 70 °C, followed by stirring for 7 h. Next, the mixture was cooled down while allowing the resin to be decanted by filtration. The system was then washed with 10% brine solution about 8 to 10 times until the pH reached 7. The yield for the ESFO was around 87%. Afterward, the epoxidized oil underwent rotary evaporation performed at 70 °C. The follow-up process consisted of the ring opening of the epoxidized sunflower oil (ESFO).

### 2.3. Synthesis of SFO Polyol

To perform the ring-opening reaction, methanol with a 7:1 molar ratio to the epoxy group was added to a three-neck round flask equipped with a condenser and dropping funnel at 65 to 70 °C. After that, 0.585 g of tetrafluoroboric acid (48 wt.% in water) was added as 0.05 wt.% of the methanol and ESFO. ESFO was added into the solution with a dropping funnel and stirred while heating for 70 min with a condenser. After the addition of the ESFO, the solution was refluxed for another 1 h in the presence of a condenser to prevent the evaporation of methanol. After the reaction was cooled down, Lewitt MP64 ion exchange resin was added to the flask and stirred to neutralize the system for about 30–45 min. The resin was filtered, and the pH of the solution was determined. Following this, rotary evaporation was performed in low and high vacuum for about 30 min and 2 h, respectively. The yield for the conversion of ESFO into SFO polyol was around 80.5%. The summarized process is shown in Figure 1.

### 2.4. Fabrication of Composite Films

A sonication method was used for the dispersion of GO in 25 mL of SFO polyol. For this, different amounts of GO or rGO (0.01, 0.02, and 0.05 wt.% based on the total weight) were dispersed in polyol for 120 min in a bath sonicator. Next, 12 mL of isocyanate was added to the prepared mixture and mechanically stirred for about 3–5 min. Then, the mixture was poured into a petri dish to cast a film. After that, a curing process was carried out at room temperature for one day and then cured at 70 °C for 90 min. Films were prepared in rectangular dimensions of 10 mm length and 1 mm width. A controlled film was also prepared using the same process without the addition of GO or rGO. The wt.% for the ingredients is presented in Table 1.

### 2.5. Characterization of the Films

Various standard techniques were used to ascertain the thermal as well as physicomechanical properties of the films. The thermal characteristics of films were studied using TA Instrument equipment (TGA Q500) with a ramp rate of 10 °C/min. The tensile strength of the films was determined using Instron 3367. MTS Qtest (SINTECH, A division of MTS Systems Corporation) was used to study the flexural behavior based on the ASTM D790 standard. The hardness of the films was measured using a PTC Instruments Type D durometer (ASTM D2240).

## 3. Results and Discussion

The presence of oxirane and hydroxyl groups was determined using the FT-IR technique. For this, a PerkinElmer Spectrum Two FT-IR Spectrometer was used to evaluate the structural changes, which were indicated by shifting or disappearing absorption peaks. The FT-IR spectra of the SFO, ESFO, and SFO polyol are presented in Figure 2a. While differentiating the SFO and ESFO spectra, it is notable that the peak at 3005 cm^−1^ which is characteristic of the C–H stretch of unsaturated carbons (H–C=C), disappeared in the ESFO spectra, which suggests the consumption of double bonds [47]. In addition, a characteristic peak of the epoxy group (C–O–C) at 832 cm^−1^ was observed in the spectrum of the ESFO, confirming the success of the epoxidation process [48]. It is noted that the C–O–C peak disappeared in the spectrum of SFO polyol, suggesting consumption of the epoxide group. Furthermore, a broad peak at around 3472 cm^−1^ appeared in the spectrum of the polyol, which is characteristic of hydroxyl stretching (O–H). This confirms the ring opening of an epoxy group to the hydroxyl group [49].

Additionally, GPC was performed to characterize the SFO, ESFO, and SFO polyol. This technique consisted of the separation of each of the components based on their differences in molecular weight. The GPC arrangement by Waters (Milford, MA, USA) was composed of four 300 × 7.8 mm phenol gel 5 µm columns. Columns presenting pore sizes of 50, 102, 103, and 104 Å were utilized. As an eluent—tetrahydrofuran (THF) was used at a 1 mL/min flow rate at 30 °C. Based on this, as shown in Figure 2b, it could be observed that SFO presented the longest elution time of around 23.2 min, which was slightly longer than the ESFO at 23.1 min, which should be expected considering that the conversion of double bonds into epoxy groups consists of the incorporation of one oxygen atom, which should lead to a slight difference in elution time. Following this, the SFO polyol had a retention time of 22.7 min, which was the fastest time in comparison to the SFO and ESFO. This reduction in retention time is an indication of an increment in molecular weight and a further confirmation of the conversion reaction from the oil to the SFO polyol.

The standard characterization techniques were used to analyze the synthesized SFO polyol. The AR 2000 dynamic stress rheometer (TA Instruments) used had a plate angle of 2° with a cone plate of 25 mm. A linearly increasing shear stress from 1 to 2000 Pa was applied at a temperature of 25 °C. The iodine value for the SFO was determined by the Hanus method (IUPAC 2.205), which was 101 g I_2_/100 g. The epoxy value (% EOC) of the ESFO was determined by the ACS PER-OXI method at 4.65%. The hydroxyl value (-OH value) of the polyol was obtained using the P.A.P. method (IUPAC 2.241), which was 191 mg KOH/g. The viscosities of the SFO, ESFO, and SFO polyol were observed to be 0.068, 0.130, and 1.61 Pa.s.

The interlayer spaces and crystal structures of the GO and rGO were determined using an XRD-6000 from Shimadzu with a Cu_α_ = 1.419 Å X-ray source. XRD was performed at a voltage of 40 kV and a current of 30 mA. Figure 3 shows the XRD patterns of the GO and rGO. One sharp peak for GO can be noticed at 2θ = 26.6°, which can be assigned to the (002) plane of graphite. This peak shifted toward a lower angle compared to the (002) plane of graphite, indicating the exfoliated nature of the GO [50]. In the rGO, a narrow peak around 2θ = 24.92° was observed, which corresponds to the (002) plane [50]. It is also worth noting that due to the diminishment of oxygenated groups from GO to rGO, there was a slight increase in the 2θ angle, along with a broadening of the peak around 24.1°. Such a difference suggests a decrease in the distance between the graphitic layers [51,52]. The absence of a strong peak in the rGO suggests an amorphous nature when compared to the GO, which presents a sharp peak at (002). In addition, the broader band at around 42.6° with a (001) orientation has been described as a turbostratic band of amorphous carbon-based materials, and there is likely to be a mixture between the sp^2^ and sp^3^ hybridization states in the carbon structure [53].

The thermal properties of the films were studied using thermogravimetric analysis (TGA) under a nitrogen atmosphere. All the tests were performed at a ramp rate of 10 °C/min. The TGA and derivative of thermogravimetric analysis (DTGA) curves for the films are provided in Figure 4. The results obtained from TGA demonstrate a single thermal transition for the control and GO-containing samples regardless of their concentration, which started around 300 °C. However, at above 600 °C, there was a higher residue percentage for the samples that contained GO. This may suggest that GO could provide a slight improvement in thermal stability, as it may prevent some of the decomposed fragments of the PU matrix to be emitted, which acted as a barrier that led to a higher percentage of residue compared to the neat film [54,55,56].

The TGA and DTG thermograms for rGO-based films are presented in Figure 5. Similarly to the previous case, there was a single thermal transition step that could suggest an absence of phase separation of the PU matrix’s segments [57]. Following this, there was a minor improvement in the thermal stability after the addition of rGO in comparison to the neat film. It is worth noting that, usually, the degradation of PU takes place in a two-step process in which the first and second thermal degradations are ascribed to the hard and soft segments, respectively, considering that there is a microphase separation that leads to this type of thermal behavior [57,58,59]. Based on the lower microphase separation of the segments in the PU matrix, it could be feasible that, upon heating, there was higher mobility of the soft segments throughout the PU chains, which promoted the dissipation of heat. This effect may have led to the single thermal transition observed in the films. Furthermore, the addition of rGO promoted the same effect as observed with the addition of GO, which led to an improvement of char residue formed at higher temperatures. Such an effect could be attributed to the formation of a thermally stable carbonaceous protective layer that prevented the decomposition of the PU matrix underneath it.

The mechanical properties were evaluated based on tensile strength, hardness, and flexural strength. Three specimens of each sample were used to perform each test to obtain an average of the values. For practicality, only the third set of samples was displayed since the difference in the values was negligible. The tensile strength over the stress behavior of the films is presented in Figure 6. The test was performed using an Instron instrument operated by the Blue Hill software system. The neat film presented a relatively tougher mechanical behavior, as it reached its yield at 22.5 MPa, which was followed by an elastic behavior due to higher elongation of around 30% until complete failure. The addition of GO into the PU matrix led to a change from tougher to more brittle mechanical behavior, similar to a fibrous structure, by presenting high tensile stress and low strain. Higher tensile stress was observed for the films containing 0.02 and 0.05 wt.% GO, as they presented tensile stress at yields of 32.5 and 26 MPa. Such behavior could be attributed to the interactions between the oxygenated groups of GO, along with the hard segments of the PU linkage due to their polarized nature [60]. However, when 0.01 wt.% GO was added, there was a considerable decrease in relation to the control sample. Along this line, it has been observed in previous studies that the incorporation of GO into a PU matrix can act as a secondary soft segment due to its two-dimensional random structure [61]. In this sense, it could be likely that the lower concentration of GO led to a disruption of the polymeric chains and promoted some microphase separation, which led to a weakening of the tensile properties. On the other hand, at higher concentrations of GO, the hydrogen bonding interactions between the oxygenated groups from GO with the aminic hydrogen from the urethane linkage may become more predominant, which was likely to promote an improvement in the tensile strength. A similar behavior was also reported in the work of Zhang et al. [60].

Based on the above, relatively lower mobility could take place, leading to more brittle behavior, as was observed. There was a decrease in the tensile strength with the addition of 0.02 to 0.05 wt.% GO. Such an effect could be due to the formation of microclusters, which decreased the overall system’s homogeneity, leading to variations in the mechanical properties throughout the film [62]. Figure 6b shows that the addition of rGO led to the opposite behavior of the tensile properties, causing the yield at the break to decrease. In this sense, the tensile strength of the initial 22.5 MPa decreased to around 12.5 MPa for both 0.01 and 0.02 wt.% rGO and to around 15 MPa for 0.05 wt.% rGO. Such behavior could have occurred due to the better interaction of rGO with the soft segments, which are more flexible and can therefore allow for more mobility of the polymeric chains [61,63,64]. The mechanical properties of the neat film went from an initially tough to more flexible behavior.

DMA is an important technique for obtaining the storage modulus and the transition temperature of materials. In this sense, these measurements were performed by using a DMA Q800 by TA Instruments with a temperature range of 30–150 °C on the specimen with a length × width × thickness of 30 × 6.5 × 2 mm, respectively. The storage modulus and loss factor (tan δ) for the GO-based films are provided in Figure 7. The DMA results show that the neat sample presented a storage modulus of around 900 MPa, which was followed by an initial decrease after the addition of 0.01 wt.% GO, leading to 600 MPa, along with increases to 1000 and 1700 Mpa for the samples that contained 0.02 and 0.05 wt.% GO, respectively. In addition, it was observed that the increase in temperature led to an overall decrease in the storage modulus, as expected from the transition from a glassy to a rubbery state. The glass transition temperature (T_g_) can be determined through the peaks to a maximum of tan δ. The measured T_g_ for the neat sample was around 55 °C. However, it was notable that the addition of 0.01 and 0.02 wt.% GO led to a slight decrease in the T_g_ to around 54 °C. On the other hand, for the addition of 0.05 wt.% GO, there was an increase in the T_g_ to 58.1 °C. Based on these results, the lower quantities of GO could have acted as secondary soft segments in the PU matrix, which may have partially disrupted its polymeric arrangement, which led to a slight decrease in the T_g_. Such behavior has been observed in previous reports [60,64]. However, when larger quantities were added, such as in the case of 0.05 wt.% GO, there was an increase in the T_g_, which could be attributed to a lamellar barrier effect, as the GO nanosheets could hinder the movement of the polymeric chains when used under relatively higher concentrations. Hence, this leads to higher required energy to allow the polymeric chains to flow [60].

Figure 8 presents the changes in the storage modulus and tan δ for the rGO-based films. Based on the results obtained from the DMA (Figure 8a), it could be observed that both the neat and 0.02 wt.% rGO films presented similar and the highest storage modulus values (900 MPa). However, the films containing 0.01 and 0.05 wt.% rGO presented relatively lower storage modulus values of around 630 and 810 MPa, respectively. Figure 8b shows the effect on the T_g_ represented by the maximum tan δ for the rGO-based films. Notably, the increasing concentration of rGO led to an overall decrease in the T_g_ in comparison to the neat sample, as the 0.01, 0.02, and 0.05 wt.% rGO T_g_ values were around 53 °C. This decrease could be attributed to the lack of oxygenated groups on rGO, which would present a weaker interaction with the hard segments, as well as possibly diminishing the crosslinking density of the polymeric structure, culminating in a decrease in the T_g_ [60].

The DSC test was performed using a DSC Q100 from TA Instruments, and the DSC graphs are presented in Figure 9. The samples demonstrated a region for the Tg that was around 50 °C. This transition matches the results observed in the DMA. Furthermore, there was no melting temperature observed, as a further increase in temperature above 250 °C would lead to the degradation of the film, as observed in the TGA data. Based on this thermal response, it was notable that the film presented a predominantly amorphous phase, which is also in accordance with the information provided by the XRD. Following this, the incorporation of GO and rGO led to negligible chances into the Tg range for the films. It is also worth noting that the absence of a crystallization temperature (T_c_) further confirms the amorphous nature of the film, and it could suggest that there was lower agglomeration activity of the GO and rGO [57].

The hardnesses of all samples were investigated by a standard test method (Shore D hardness). The measurement was performed three times for each sample, and Figure 10 shows their average hardness. Based on the results obtained, it was noticed that there was a continuous increase in hardness with increasing additions of rGO followed by the addition of GO. The neat film, 0.05 rGO, and 0.05 wt.% GO presented hardness values of 51, 60, and 67 Shore D, respectively. These results provide a standard deviation of 5.79. The improvement observed with increasing amounts of rGO could be attributed to the interlinkage of the rGO nanosheets, which could interact with the soft segments due to similar polarity. On the other hand, a further increase in hardness could be observed with the incorporation of GO due to the presence of oxygenated groups that are more likely to form stronger interactions with the urethane linkages, which led to increments in hardness values [31,65].

Three-point bending was performed to analyze the flexural properties of the films (Figure 11). In the case of the GO-based films (Figure 11a), it could be observed that there was an improvement in the flexural strength upon the addition of 0.02 wt.% GO, along with a slight increment in extension, of 5 N and 8 mm, respectively. However, the further addition of 0.05 wt.% GO presented a decrease in lower flexural strength, which was the same as the neat sample of 2.25 N. Following this, 0.01 wt.% GO presented the lowest flexural strength of 0.2 N and the highest extension of 8.7 mm. In the case of the rGO-based films (Figure 11b), it could be observed that the addition of 0.02 wt.% rGO yielded a considerable improvement in the flexural strength, which reached around 14 N, along with an extension of 10.5 mm. On the other hand, the addition of 0.01 and 0.05 wt.% rGO led to a decrease in this property when compared to the neat sample.

The hydrophobicity of the film was analyzed through the water contact angle (WCA) performed using the Ossila Contact Angle Goniometer. For this, a droplet of 10 µL distilled water was set on the surface of the testing sample. The water contact angles were recorded and are shown in Figure 12. Based on this, it could be noted that there was an overall increase in the contact angle after the addition of either GO or rGO in comparison with the neat sample, which presented 78.15°. The highest angle reached by the films containing rGO was the one containing 0.02 wt.%, which reached 100.20°. Following this, the 0.02 wt.% GO reached a contact angle of 81.92°. The standard deviation for the water contact angles was 10.63. The expected increase in hydrophobicity can be attributed to the graphitic structure present in both GO and rGO. In addition, the contact angle of the GO was lower when compared to the rGO due to the larger presence of oxygenated groups in its structure.

Lastly, tapping-mode atomic force microscopy (AFM) was performed to observe the effects of the incorporation of GO and rGO into the PU film’s surface, and the images for the BPU films are displayed in Figure 13. Through this, it could be seen that the neat film presented a relatively rougher surface when compared to the films containing GO and rGO. Despite this, as the concentrations of GO and rGO increased, there were also mild increases in their respective surface roughnesses.

## 4. Conclusions

This work successfully proposes the synthesis of a sunflower oil-based polyol through a two-step process of epoxidation followed by ring-opening. This process promoted the chemical conversion of double bonds into hydroxyl groups in the sunflower oil, turning it into a suitable polyol. The polyol was used to make PU films, which were composited through simple blending with different amounts of GO and rGO. The mechanical properties in terms of tensile strength showed an improvement from 22.5 to 32.5 MPa comparing the neat sample and that with 0.02 wt.% GO, respectively. However, there was a decrease in the tensile strength after the incorporation of rGO. A similar trend was observed for the storage modulus, as the neat sample presented a value of 900 MPa, whereas the 0.02 and 0.05 wt.% GO samples presented 1000 and 1700 MPa, respectively. There was also a general decrease in the storage modulus after the addition of different concentrations of rGO. In addition, there was a continuous increase in hardness, which went from 51 to 60 Shore D when comparing the neat and 0.05 wt.% of GO. During the thermal studies, it was observed that both GO and rGO promoted negligible variations in the film’s thermal properties. Lastly, based on the obtained results, it could be suggested that the improvement in properties based on the addition of GO rather than rGO could be related to the stronger interactions between the oxygenated groups from GO with the PU’s hard segments in comparison to the weaker interactions of rGO with the aliphatic soft segments. Thus, this work demonstrates a viable fabrication process of a sunflower oil-based PU film that presented a relatively low concentration of GO or rGO, along with appreciable properties, which can make it a suitable material for coating applications that require mechanical protection over a substrate in terms of impact, tensile strength, and hardness. Furthermore, the composite coating can be suitable for applications related to coatings that require a hydrophobic nature as well as chemical stability under aggressive environments.

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
