# Peer review of "Effect of Graphene Oxide and Reduced Graphene Oxide on the Properties of Sunflower Oil-Based Polyurethane Films"

_polymers, 2022, doi:10.3390/polym14224974_

Round 1

Reviewer 1 Report

In this work, the sunflower oil-based polyurethane films assisted by graphene oxide and reduced graphene oxide were fabricated. And their chemical composition and mechanical properties were discussed in detail. However, there are still some issues to be addressed:

1. There are some grammar and format mistakes. Please check the whole manuscript carefully.

2. In the Introduction part,

a) Why selects the sunflower oil? why selects the GO and rGO?

b) The novelty of your work needs to be further emphasized.

3. The details of characterization tests and instrument information need to be completed, e.g., FTIR, XRD, water contact angle.

4. What are the application fields of your work?

Author Response

All the comments were addressed. Please find attached our response to the comments.

Reviewer 2 Report

In this manuscript, the authors synthesized a sunflower oil-based polyol through a two-step process of epoxidation followed by ring-opening, which was subsequently used for the preparation of polyurethane films as well as composites blended with (reduced) graphene oxides (GO/rGO). Through systematic investigations on their thermal/mechanical performance, the authors found that both GO and rGO have some impacts on the properties of polyurethane, such as increased tensile strength, improved storage modulus and enhanced hardness when appropriate amount of GO was introduced. Overall, the studies here could provide some useful information for the reader in the related areas. The following are specific comments/questions that need the authors’ attention:

1.     In the Results and Discussion part regarding the synthesis of SFO, ESFO and SFO polyol, such as data in Figure 2, why the authors didn’t use 1H NMR to characterize them, which is a very commonly used and reliable technique.

2.      Page 6, line 189: “However, at higher temperatures, there was a higher residue percentage for the samples that contained GO…”, the authors need to be more specific on the temperature mentioned here (the specific temperature value), otherwise, it is difficult for the readers to figure out which temperature range the authors were talking about.

3.      The authors should provide the differential scanning calorimetry (DSC) data to show the thermal properties of the studied polyurethanes as well as the effects after introducing GO/rGO.

4.      For stress-strain studies, at least 3 runs on each sample should be performed and the authors are suggested to use the average values for the comparison of mechanical performance.

5.      Page 7, line 221: “Such behavior could be attributed to the interactions between the oxygenated groups of GO along with the hard segments of the PU linkage due to their polarized nature”, if what the authors claimed here is true, then why the stress from the sample with 0.01 GO incorporated (Figure 6a, red curve) showed a lower value?

6.      Page 8, line 250: “Based on these results, the lower quantities of GO could have acted as secondary soft segments in the PU matrix which may have partially disrupted its polymeric arrangement and therefore decreased its Tg”, could the authors provide related reference for this claim?

Author Response

(The authors gave the same response as above.)

Reviewer 3 Report

1) The optical photos of the films show that the graphene distribution is inhomogeneous. How did the authors try to mitigate agglomeration of graphene?

2) The experimental descriptions should be more precise. The wt% for GO was reported based on the total weight including the solution, or only the weight of the SFO polyols. Add a table with wt% values for all ingredients so they add up, and easy to follow and understand.

3) The novelty and potential impact of the work should be better presented in the manuscript. In its current form these aspects are unclear.

4) The isolated yield for both the synthesis of epoxidized sunflower oil and the synthesis of SFO polyol should be reported. Include spectra and chromatogram for purity demonstration.

5) The authors introduce biobased polyols, where the diverse and emerging applications of these polyols in polymer science should be exemplified (10.1021/acsapm.1c01081; 10.1039/D0GC03226C; 10.1039/D1GC00445J).

6) The reproducibility of the results should be demonstrated. Therefore, add error bars / standard deviations for the hardness values in Figure 9, water contact angles in Figure 12. Independently prepared samples should be used to obtain the standard deviations.

7) SEM images should be provided for cross-section as well as surface analysis at different magnifications to reveal the morphology of the films. SEM analysis of any films is reported in the literature is crucial.

8) AFM analysis should be performed and reported for the films.

9) The recycling of the materials should be discussed. What is the end-of-life disposal of these materials? How can they be recycled and reused?

10) GO/rGO were shown to significantly alter the properties of various films, which should be briefly listed (10.1021/acsapm.1c00978; 10.1016/j.mtchem.2021.100602; 10.1016/j.cej.2021.129763).

11) The purity and grade of all materials, chemicals and solvents should be mentioned under the Materials section of the manuscript for reproducibility purposes.

12) Figure 11 should be deleted as it is replicated in Figure 12.

Author Response

(The authors gave the same response as above.)

Round 2

Reviewer 1 Report

I am satisfied with the revised manuscript.

Reviewer 2 Report

Can be considered for publication for this revised manuscript

Reviewer 3 Report

Can be published.